# Computational Identification of Potential Multitarget Inhibitors of Nipah Virus by Molecular Docking and Molecular Dynamics

**DOI:** 10.3390/microorganisms10061181

**Published:** 2022-06-09

**Authors:** Vinay Randhawa, Shivalika Pathania, Manoj Kumar

**Affiliations:** 1Virology Discovery Unit and Bioinformatics Centre, CSIR-Institute of Microbial Technology, Council of Scientific and Industrial Research, Chandigarh 160036, India; vinay.plp@gmail.com (V.R.); shivalika20@gmail.com (S.P.); 2Academy of Scientific and Innovative Research (AcSIR), Ghaziabad 201002, India

**Keywords:** Nipah virus, molecular docking, small-molecule inhibitors, molecular dynamics, drug repurposing, multitarget inhibitor

## Abstract

Nipah virus (NiV) is a recently emerged paramyxovirus that causes severe encephalitis and respiratory diseases in humans. Despite the severe pathogenicity of this virus and its pandemic potential, not even a single type of molecular therapeutics has been approved for human use. Considering the role of NiV attachment glycoprotein G (NiV-G), fusion glycoprotein (NiV-F), and nucleoprotein (NiV-N) in virus replication and spread, these are the most attractive targets for anti-NiV drug discovery. Therefore, to prospect for potential multitarget chemical/phytochemical inhibitor(s) against NiV, a sequential molecular docking and molecular-dynamics-based approach was implemented by simultaneously targeting NiV-G, NiV-F, and NiV-N. Information on potential NiV inhibitors was compiled from the literature, and their 3D structures were drawn manually, while the information and 3D structures of phytochemicals were retrieved from the established structural databases. Molecules were docked against NiV-G (PDB ID:2VSM), NiV-F (PDB ID:5EVM), and NiV-N (PDB ID:4CO6) and then prioritized based on (1) strong protein-binding affinity, (2) interactions with critically important binding-site residues, (3) ADME and pharmacokinetic properties, and (4) structural stability within the binding site. The molecules that bind to all the three viral proteins (NiV-G ∩ NiV-F ∩ NiV-N) were considered multitarget inhibitors. This study identified phytochemical molecules RASE0125 (17-O-Acetyl-nortetraphyllicine) and CARS0358 (NA) as distinct multitarget inhibitors of all three viral proteins, and chemical molecule ND_nw_193 (RSV604) as an inhibitor of NiV-G and NiV-N. We expect the identified compounds to be potential candidates for in vitro and in vivo antiviral studies, followed by clinical treatment of NiV.

## 1. Introduction

Nipah virus (NiV), a virus of the genus Henipavirus within the Paramyxoviridae family, was initially isolated in 1999 during an outbreak in Malaysia [1]. Since then, there have been sporadic reports of NiV outbreaks in various other Asian countries such as Singapore, the Philippines, Bangladesh, and India, the most recent of which is in the South Indian state of Kerala [2]. NiV is potentially pandemic, highly pathogenic, and capable of infecting most mammalian species. From 1998 to 2015, more than 600 cases of NiV human infections were reported, with a fatality rate of 38% in Malaysia and 43–100% in India and Bangladesh [2].

NiV is classified as a biosafety level 4 (BSL4) pathogen due to its high case fatality rate following human infection and due to the lack of effective therapeutics or vaccines for infection treatment [3]. There have been some efforts to manage NiV by searching for small-molecule therapeutics [4,5]. For example, favipiravir has demonstrated efficacy against a broad spectrum of RNA viruses such as Paramyxoviridae, Filoviridae, Arenaviridae, and Bunyavirales; however, a recent in vitro study also demonstrated favipiravir to be inhibiting NiV and Hendra virus (HeV) virus replication and transcription at micromolar concentrations [6]. R1479 (4′-azidocytidine), a drug previously identified to inhibit flaviviruses, is also found to inhibit henipaviruses, including other paramyxoviruses, with high potency [7]. Small molecules that activate IRF3 and modulate RIG-I-like receptors pathways have also been investigated as potential strategies for targeting NiV infection [8]. However, despite these efforts for developing small-molecule inhibitors, no approved drugs are available yet for efficient use in humans. Though Ribavirin is not a proven treatment for NiV, it is particularly being used in a state of emergency to treat acute NiV encephalitis as a first-line treatment strategy [9]; however, it offers various side effects such as nausea, vomiting, and convulsions during NiV treatment [10]. Additionally, driven by the unwanted pharmacokinetics and numerous side effects of synthetic compounds, there is also a specific interest in using compounds of natural origin (phytochemicals) to overcome the side effects. Therefore, taken altogether, there is a strong need to discover novel small-molecule chemical inhibitors or phytochemicals against NiV with potential therapeutic value and fewer or no side effects in humans.

Contrary to conventional drug discovery approaches that rely on targeting a single protein, the next paradigm in drug discovery is to search for multitarget drugs. Viral entry into the host begins with relatively nonspecific interactions between the virus and attachment factors on the cell surface. The henipaviruses encode two envelope glycoproteins, including G and F glycoproteins. These virus attachment glycoproteins are essential for the recognition of host-cell-surface receptors ephrin-B2 (EFNB2) and ephrin-B3 (EFNB3), which mediate cellular attachment, fusion, and virus entry [11]. NiV glycoprotein G (NiV-G) has a globular head domain formed of a six-bladed beta-sheet propeller, connected via a flexible stalk domain to a transmembrane anchor. The binding of G glycoprotein to its ephrin receptors leads to conformational changes in glycoprotein, followed by subsequent conformational changes in F glycoprotein (NiV-F) that lead to the establishment of a physical link between viral and cellular membranes by stripping of the fusion peptide. Concluding the fusion process, nucleocapsid enters the host cell cytoplasm, resulting in the onset of viral replication [12]. The NiV RNA genome is encapsidated by the nucleoprotein (NiV-N) that protects the viral genome from degradation and also acts as a template for mRNA transcription [13]. Considering the role of each of the three viral proteins (i.e., NiV-G, NiV-F, and NiV-N) in virus replication and spread, they are, therefore, of particular interest in anti-Nipah drug discovery.

Multitarget drug discovery has raised considerable interest in the last decade [14,15,16,17]. Most drugs aim at a single biological molecule, widely known as the “one target, one drug” strategy, whereas a multitarget drug has the potential to modulate the effect of multiple targets. Multitarget drugs have even been approved for clinical use [18], and many complex diseases, including neurodegenerative diseases, cardiovascular diseases, and cancers, are often treated with multidrug therapy or a “cocktail” of drugs [19]. Many computational approaches have been developed to address polypharmacology-guided drug discovery [16,20]; these approaches, including virtual screening, molecular docking, and molecular dynamics, have been widely implemented in modern drug discovery and are capable of screening new compounds with multitarget characteristics. For example, docking simulations have identified a series of novel multitarget compounds (e.g., donepezil–indolyl hybrid, donepezil–pyridyl hybrid, donepezil hybrid, etc.) for the treatment of neurodegenerative diseases [15]. In another study, Zhou et al. (2017) [21] implemented a computational drug design method that aided the synthesis and characterization of a novel multikinase inhibitor molecule. Yousuf et al. (2017) [22] proposed novel multitarget inhibitors for breast cancer treatment by using a virtual screening/molecular docking-based approach. Using multitargeted molecular docking Singh and Bast (2015) [23] also identified various potential natural compounds as inhibitors of tyrosine kinase receptors involved in the development of several types of cancers. Computational approaches for multitarget drug discovery in pathogens have also been reported [24,25,26]. Several studies have been reported in the literature addressing various computational methods (e.g., virtual screening, molecular docking, molecular modeling, machine learning, quantitative structure–activity relationship, in silico ADME, pharmacophore, etc.) targeting multiple targets [27]. 

Our group has already developed various therapeutic/antiviral resources [28,29] and methods to predict antiviral peptides [30,31], small molecules [32,33], and siRNAs [34]. Our group has developed a resource of multitargeted putative therapeutics and epitopes for NiV drug discovery as well [35]. Additionally, several small-molecule inhibitors were identified by our group using QSAR [36] and molecular docking-based studies [37]. In our previous study [37], we have identified small-molecule FDA-approved drugs as potential inhibitors of NiV-G using an ensemble of molecular docking and analysis of the chemical–protein interaction network. Computational screening of small-molecule inhibitors/drugs is a useful approach for rapid screening of therapeutic molecules from a large chemical space; this motivated us to shortlist potential leads to meet the urgent demand for repurposing drugs for the treatment of NiV. In this study, to extend the scope of already developed methods and approaches against NiV, an integrative structure-based approach was implemented to identify potential multitarget small-molecule chemical/phytochemical inhibitor(s) against NiV-G, NiV-F, and NiV-N—the three most important NiV proteins targets. The workflow for the proposed integrated approach for multitarget molecule screening is presented in Figure 1.

## 2. Materials and Methods

### 2.1. Data Compilation and Small-Molecule Library Preparation

An extensive literature search in PubMed (pubmed.ncbi.nlm.nih.gov/) database was performed to extract the structural information on known NiV inhibitors using the following integrated text query: (((Nipah) AND drug*)) OR ((Nipah) AND inhibit*). Then, 3D chemical structures of small-molecule NiV inhibitors were prepared and edited using MarvinSketch v5.10.0 software (https://chemaxon.com/). To probe for NiV inhibitors that might be associated with adverse side effects, ADME and pharmacokinetic analyses were performed using the SwissADME webserver (http://swissadme.ch/) (accessed on 18 October 2021).

Data on different aspects of phytochemicals, including their 3D structures, were obtained from three well-established databases: (1) SerpentinaDB [38], for a structured compilation of Indian snakeroot (*Rauvolfia serpentina*) plant-derived molecules (*n* = 142); (2) Phytochemica [39], for a structured compilation of molecules from the plants *Atropa belladonna*, *Catharanthus roseus*, *Heliotropium indicum*, *Picrorhiza kurroa*, and *Podophyllum hexandrum* (*n* = 868); (3) Phytochemical and Drug Target DataBase (PDTDB, *n* = 221) [40]; these databases contain a vast amount of information on phytochemicals having therapeutic potential against various diseases. All obtained chemical structures were structurally optimized via energy minimization (500 steps of steepest descent) with Merck Molecular Force Field (MMFF94) while implementing OpenBabel v 2.4.0 software [41].

### 2.2. Chemical Variability Analysis of Compiled Molecules

The R-ChemmineR package [42] was used to cluster the literature-compiled NiV inhibitors into their discrete similarity groups (on the basis of the Tanimoto similarity measure/index), using fingerprints generated from the descriptor vectors. A maximum common substructure search was also performed to identify the potential representative scaffolds in the inhibitors using the flexible common substructure (FMCS) algorithm [43].

### 2.3. Retrieval and Assessment of Viral Protein Structures

The 3D structures of Nipah glycoprotein (NiV-G), fusion protein (NiV-F), and nucleoprotein (NiV-N) were obtained from the RCSB Protein Data Bank (PDB) (https://www.rcsb.org/) (accessed on 10 October 2021) having identifiers 2VSM (1.8 Å) [11], 5EVM (3.3 Å) [44], and 4CO6 (2.5 Å) [45], respectively. The 3D structures of proteins were assessed in Chimera v 1.16 (http://cgl.ucsf.edu/chimera/), and any missing residue(s) was/were fixed, and optimized. The structure of NiV-N (4CO6) is a NiV nucleoprotein–phosphoprotein complex, and therefore, the phosphoprotein chains were removed from the complex. Nucleoprotein structure also comprised many modified residues (e.g., MSE/AMSE/BMSE), which were modified to Gly, the simplest amino acid, in Pymol v 2.3.0 (https://pymol.org/2/) software. The modified protein structure was structurally optimized via energy minimization (1000 steps of steepest descent) in Chimera.

### 2.4. Assessment of Ligand-Binding Pockets

The data on ligand-binding sites in all three viral proteins were obtained from the published literature. The binding-site residues of protein NiV-G (Gln559, Glu579, Tyr581, Ile588) were obtained from Kalbhor et al. (2021) [46], while binding-site residues of NiV-F (His29, Tyr30, Val39, Lys40, Asn380, Tyr432, Leu433) and NiV-N (Lys34, Arg36, Phe38, Val58, Ala65, Ser67, Glu124, Leu128, Ile131) were obtained from Sen et al. (2019) [47]. To assess the conformational/structural differences among the ligand-binding pockets in the viral protein structures, their respective pocket volume sizes were assessed using Pocket Volume Measurer (POVME) [48] and Pymol software.

### 2.5. Molecular Docking

For molecular docking, the cleaned and energy-minimized structures (called receptors henceforth) were considered (see Section 2.3). The three-step docking comprises the following preparations: (1) receptor preparation—using Autodock Tools 4.2.6 software [49], polar hydrogen atoms were added, and the protonation states were assigned by including appropriate Gasteiger charges to the respective protein models (NiV-G, 8.9803; NiV-F, 2.986; NiV-N,0.9779). All other receptor preparation options were kept at default; (2) ligand preparation—charges were added to small molecules (chemical inhibitors and phytochemicals), and all bonds of ligands were set to be rotatable except N–C bonds; (3) molecular docking—all ligands were docked into the respective ligand-binding sites using the QuickVina v 2.0 [50] software, a docking tool that accelerates AutoDock Vina software [51] by implementing an already benchmarked (on CCDC/ASTEX dataset) molecular docking approach [37]. The small molecules that bind to all the three viral proteins (NiV-G ∩ NiV-F ∩ NiV-N) were considered multitarget inhibitors. Intermolecular interactions among receptors and ligands were analyzed with LigPlot+ v 2.2 software (https://www.ebi.ac.uk/thornton-srv/software/LigPlus/), and 3D images were rendered with Pymol.

As a positive control, molecular docking studies were also performed among viral proteins and the drug remdesivir (GS-5734) [52]. The 2D structure of remdesivir (DB14761) was obtained from the DrugBank database (https://www.drugbank.com/) (accessed on 18 April 2022), which was 3D optimized (using the obminimize module), with 500 steps of conjugate gradient and steepest descent methods. Remdesivir was then docked into the respective binding sites of all three viral proteins using the implemented docking pipeline.

### 2.6. Computing ADME and Pharmacokinetics Properties of Small Molecules

ADME and pharmacokinetics properties of docking-based prioritized molecules were computed using ADMETlab 2.0 webserver [53]. ADMETlab computes a total of 88 molecular property descriptors based on molecule’s physicochemical properties (*n* = 17), medicinal chemistry (*n* = 13), and ADMET properties [A (*n* = 7), D (*n* = 4), M (*n* = 10), E (*n* = 2), T (*n* = 35)]. To prioritize these molecules for next step, the respective z-scores for each molecule were computed based on ADME descriptors. Briefly, the most relevant property value values $x_{i}\;\left(x_{1},\ldots ,x_{n}\right)$ were first transformed into a data matrix of binary variables (0 or 1). During data transformation, the values within the standard limits were transformed into binary variable 1, else 0. All $x_{i}$ values were then transformed into a normalized score $z_{i}$ for each molecule, which was computed as z=(X–μ)/σ, where *X* is the value being analyzed, μ is the mean, and σ is the standard deviation. In addition, z-scores for molecules were computed in R v 3.6.3 (https://www.r-project.org/) statistical environment.

### 2.7. Assessing Gene Expression Induction by Molecules

Based on the positive z-scores, the following molecules were selected: ND_nw_193 (RSV604), CARS0358 (NA), and RASE0125 (17-O-Acetyl-nortetraphyllicine). The potential up- and downregulated human protein targets of these molecules were predicted based on their predicted pharmacological activity (Pa (activity probability) > 0.6) using the DIGEP-Pred server [54]. DIGEP-Pred uses the prediction of activity spectra for substances (PASS) algorithm to estimate various kinds of biological activities (including the interaction of small molecules with gene/proteins) by querying the 2D structure with those of well-known biologically active substances (i.e., activity = f (structure)). PASS prediction results are represented by the list of activities with probabilities ‘‘to be active’’ Pa and ‘‘to be inactive’’ Pi. The background data (i.e., drug-induced changes in gene expression) for DIGEP-Pred was taken from the Comparative Toxicogenomics Database (CTD) [55], which provides per se general drug-induced changes in gene expression [56], and such changes in gene expression represent the intrinsic properties of a drug molecule.

### 2.8. Molecular Dynamics Simulations

Molecular dynamics (MD) simulations were performed to ascertain the structural binding stability of 3 potential multitarget NiV inhibitors—RASE0125, ND_nw_193, and CARS0358—with respective viral protein structures (i.e., NiV-G-ND_nw_193 complex, NiV-G-CARS0358 complex, and NiV-G-RASE0125 complex; NiV-F-ND_nw_193 complex, NiV-F-CARS0358 complex, and NiV-F-RASE0125 complex; NiV-N-ND_nw_193 complex, NiV-N-CARS0358 complex, and NiV-N-RASE0125 complex). As respective controls, MD simulations were also performed for free protein structures (apo-NiV-G; apo-NiV-N; apo-NiV-N). In total, 12 MD simulations were performed. Groningen Machine for Chemical Simulation (GROMACS) v 2020.1 software [57] was used for simulations utilizing CHARMM36 force field v 2019 [58] at 300 K. The topology parameters for inhibitor were produced in the CGenFF server (http://cgenff.umaryland.edu/) (accessed on 11 April 2022) and complexed into the protein topologies to make protein–inhibitor complexes. Systems were solvated using the simple point charge water model (spc216) in cubic boxes and counter ions (Na^+^ or Cl^−^) were added to neutralize the systems. Solvated systems were then minimized with 1000 steps using the steepest descent method, followed by the equilibration run (NVT + NPT), for 100 picoseconds. After equilibration, a 5 ns production simulation (MD run) was performed for all the 12 established systems (i.e., 9 complexes and 3 free proteins). The trajectories were analyzed by assessing C-α root-mean-square deviations (RMSDs) and root-mean-square fluctuation (RMSF) using “gmx rmsd” and “gmx rmsf” modules, respectively. The trajectories were plotted gnuplot-x11 program (http://gnuplot.info/).

### 2.9. Principal Component Analysis and Gibbs Free Energy Calculation

To explore the conformational changes docked molecules—RASE0125, ND_nw_193, CARS0358—brought into the viral protein structures, principal component analysis (PCA) was carried out utilizing the essential dynamics approach. Using the GROMACS “gmx anaeig” and “gmx covar” modules, the covariance matrix was calculated (considering the protein backbone atoms), followed by the diagonalization to calculate the eigenvectors and eigenvalues. Principal component analysis (PCA) was calculated from the trajectory, and the first two components (PC1 and PC2) were used for the plotting. The Gibbs free energy landscapes (FELs) were developed using the “g_sham” module to capture the lowest energy stable state. The first two principal components were used to calculate the FEL based on the equation $\Delta G\left(PC1,PC2\right)=-K_{B}TlnP\left(PC1,PC2\right)$. PC1 and PC2 are reaction coordinates, KB symbolizes the Boltzmann constant, and P (PC1, PC2) illustrates the probability distribution of the system over the first two principal components.

## 3. Results and Discussion

### 3.1. Compiling Small-Molecule Inhibitors of NiV 

An extensive literature search was performed to obtain information on small-molecule NiV inhibitors or drug molecules. The PubMed database was queried by combining suitable keywords and Boolean operators to first obtain relevant articles. Next, these articles were manually curated to archive molecules’ data and their additional details, including chemical name, IUPAC name, 2D structure, references, etc. Finally, information on a total of 206 NiV inhibitors was compiled (Appendix A).

To assess the diversity and unique molecular scaffolds that were highly prevalent in compiled NiV inhibitors, these were clustered into their discrete groups on the basis of the Tanimoto similarity measure [59], one of the most widely used similarity measures for comparing chemical structures in cheminformatics. Clustering results were visualized by inspecting the multidimensional scaling (MDS) plot that indicated 206 inhibitors to be broadly grouped into four distinct clusters: Cluster 1 (*n* = 57), Cluster 2 (*n* = 35), Cluster 3 (*n* = 16), and Cluster 4 (*n* = 7). The molecules in each of the clusters comprised compounds with similar structural patterns (Figure 2), while the rest of the 91 molecules were present as independent entries.

#### ADME and Pharmacokinetic Properties of Small Molecules

To probe for NiV inhibitors that might be associated with adverse side effects, ADME and pharmacokinetic properties of literature-compiled small-molecule inhibitors were computed using the SwissADME webserver (Appendix A). SwissADME uses comparative information derived from poorly and highly absorbed drugs to predict passive intestinal absorption and brain penetration, as a function of lipophilicity and apparent polarity (described by WLOGP and TPSA, respectively). For all compiled inhibitors, the predictions were displayed as a bi-plot, also called the BOILED-Egg plot [60] (Figure 2). A total of 160 molecules (~78%) fall inside the whole bi-plot, where white and yellow ellipses indicate molecules having a high probability of good intestinal absorption and blood–brain barrier (BBB) crossing, respectively. A total of 46 molecules (~22%) were predicted as not absorbed by the GI and BBB non-permeant, which are located in the grey area of the plot; these molecules were explicitly excluded from our analyses, as they are likely associated with side effects. The ellipsoidal regions defining the intestine and brain permeation were drawn as initially reported. The selected molecules were also analyzed for drug-relevant properties on the basis of “Lipinski’s rule of five”, which indicated a total of 149 (72%) to have drug-like properties.

### 3.2. Compiling Information on Small-Molecule Phytochemicals

Phytochemicals are plant-derived small-molecule inhibitors, which are rich sources of diverse scaffolds that could serve as a basis for rational drug design. Many plants have shown effective antiviral and immune-boosting potentials against viruses including SARS-CoV, Zika, Ebola, Nipah, and other highly pathogenic viruses [61]. Therefore, we also prospected for potential phytochemical molecules against NiV proteins. Information on different features of phytochemicals, including their 3D structures, was obtained from three well-established small-molecule databases—namely, SerpentinaDB (*n* = 142), Phytochemica (*n* = 868), and Phytochemical and Drug Target DataBase (PDTDB, *n* = 221); these databases contain extensive information on phytochemicals having therapeutic potential against various diseases. The molecules in these databases already have information on pharmacokinetic properties and, therefore, were not computed in our analyses. 

### 3.3. Volumetric Analyses of Ligand-Binding Protein Pockets

The data on ligand-binding sites in all three viral proteins were obtained from the published literature. The binding-site residues of protein NiV-G (Gln559, Glu579, Tyr581, Ile588) were obtained from Kalbhor et al. (2021) [46]. The residues Ile588 and Tyr581 participate in hydrophobic interactions and contribute to the binding pocket that accommodates Phe120 of human EFNB2; hence, they are considered critical for inhibiting interactions by which NiV attach to EFN receptors [62]. The binding-site residues of NiV-F (His29, Tyr30, Val39, Lys40, Asn380, Tyr432, and Leu433) and NiV-N (Lys34, Arg36, Phe38, Val58, Ala65, Ser67, Glu124, Leu128, and Ile131) were obtained from Sen et al. (2019) [47]. Pocket volumes of NiV-F, NiV-G, and Niv-N were 69 Å³, 23 Å³, and 60 Å³, respectively; this indicated availability of similar conformational space for ligands in NiV-F and Niv-N but a smaller space for NiV-G.

### 3.4. Molecular Docking

Docking studies were carried out in order to find optimal conformations of all ligands into the respective binding pockets of NiV-G, NiV-F, and NiV-N. As a post-docking filter, molecules with the best binding affinity (≤−7.0 Kcal/mol) were considered, which predicted 300, 827, 685 molecules to be docking inside NiV-N, NiV-G, and NiV-F, respectively. The molecules were then shortlisted based on their interaction with the binding-site residues. Of 300 molecules in NiV-N, 227 molecules bind to ligand-binding sites (Lys34, Arg36, Phe38, Val58, Ala65, Ser67, Glu124, Leu128, and Ile131), whereas of 827 molecules in NiV-G, 620 molecules bind to the ligand-binding sites (Gln559, Glu579, Tyr581, and Ile588). Similarly, of 685 molecules in NiV-F, 681 molecules bind to ligand-binding sites (His29, Tyr30, Val39, Lys40, Asn380, Tyr432, Leu433). 

To gain an understanding of differences in the number of docked small-molecule inhibitors, volumetric differences among ligand-binding pockets of proteins were analyzed. The difference in binding of a comparatively larger number of small molecules in NiV-F and NiV-N could be attributed to a comparatively wide open and large pocket, compared with NiV-G. The results are also contrary to the pocket volumes of viral proteins (Niv-G, 23 Å³; NiV-F, 69 Å³; Niv-N, 60 Å³). The inhibitor molecules that bind to all three viral proteins (NiV-G ∩ NiV-F ∩ NiV-N) were considered. Overall, based on binding energy threshold (≤−7.0 Kcal/mol), interactions with ligand-binding-site residues, a total of 156 molecules are selected as common inhibitors of NiV-G, NiV-F, and NiV-N. Since 156 was a large number for further analyses, the binding-energy threshold was raised to −7.8 Kcal/mol, which predicted a comparatively smaller number of lists as common inhibitors of three viral proteins. Based on the readjusted threshold, a total of eight molecules were selected, of which two were chemical inhibitors, and six were phytochemicals. The binding affinities of eight molecules within the ligand-binding pocket are provided in Table 1, and the information on intermolecular interactions, including hydrogen bonding and hydrophobic interactions, is provided in Appendix A. Chemical details of molecules, including IUPAC and SMILES, are provided in Appendix A.

As a positive control, molecular docking studies were also performed with the drug remdesivir (GS-5734). Remdesivir is a nucleotide prodrug that has broad antiviral activity against viruses from different families and therapeutic efficacy in nonhuman primate models of lethal Nipah virus infection [63] and is, therefore, likely effective against Nipah virus infection in humans [53]. Remdesivir was also found to interact with all three viral proteins but with comparatively low binding affinity. Remdesivir binds to NiV-G with a binding affinity of −7.5 Kcal/mol, and it binds to NiV-F (−6.6 Kcal/mol) and Niv-N (−5.6 Kcal/mol) with comparatively poor affinity, which is attributed to weak intermolecular interactions among remdesivir and proteins.

### 3.5. Computing ADME Properties for Molecule Prioritization

ADME and pharmacokinetic properties of docking-based prioritized molecules were computed using the ADMETlab webserver. The pharmacokinetic properties were transformed into z-scores and considered for further molecule prioritization (Appendix A). A positive z-score indicates molecules that have more drug-like properties, and therefore, molecules were ranked and prioritized based on positive z-scores; this threshold shortlisted a total of three molecules: ND_nw_193 (RSV604), CARS0358 (NA), and RASE0125 (17-O-acetyl-nortetraphyllicine) (Figure 3). Information on ADME-based z-score is provided in Table 2. The docked conformations of the three molecules into their respective ligand-binding sites are provided in Figure 4.

### 3.6. Assessment of Gene Expression Induction by Small Molecules

Drug molecules interact with many off-targets that could lead to various side effects [64]. In fact, interactions between drug molecules and their targets/off-targets can often induce changes in expression profiles of many genes/proteins [65] that can either result in activation, repression, or dysregulation of downstream signaling pathways [66]. Therefore, to prioritize the molecules further, only those molecules were considered that could lead to dysregulation of expression profiles of a comparatively smaller number of human proteins. The potential up- and downregulated human protein targets of molecules—RASE0125, ND_nw_193, CARS0358—were predicted based on their predicted pharmacological activity (Pa (activity probability) > 0.6) using the DIGEP-Pred. It was observed that molecule RASE0125 could upregulate gene expression of proteins KRT18 (Pa = 0.670) and RAC1 (Pa = 0.621), while it downregulated CHEK1 (Pa = 0.655). Similarly, ND_nw_193 could only upregulate gene expression of ATG5 (Pa = 0.670). CARS0358 was not found to be regulating the expression profiles of any of the proteins. Overall, these three small-molecule inhibitors could induce the expression profiles of only a fewer number of proteins that could ultimately avoid dysregulation of critically important signaling pathways, thus reducing the risk of drug-induced severe side effects.

### 3.7. Molecular Dynamics Simulations

MD simulation presents an approach for the structural refinement of docked complexes [67]. In order to refine and examine the stability of the three selected molecules (RASE0125, ND_nw_193, and CARS0358), MD simulations of corresponding protein–ligand complexes, as well as free proteins, were performed for 5 ns. Since remdesivir, the positive control, was not able to well bind to all of the three viral protein targets, we did not perform its MD simulations. The stability of complexes was assessed by computing two main structural parameters: Cα-RMSD and -RMSF. RMSD is a standard measure of computing structural distance between coordinates and measures the average distance between a group of atoms. As a control, Cα RMSD values of all free proteins were also computed. Complexes of molecules RASE0125 and CARS0358 were found to be stable in all the three viral protein structures and plateaued to average RMSDs of ~0.12 nm, ~0.35 nm, and ~0.12 nm in NiV-G, NiV-F, and NiV-N protein structures (Figure 5). It was interesting to observe that the docked complexes were more stable than the free proteins. Apart from protein complexes, the RMSDs of bound molecules were also computed during the simulations, and both the molecules were found to be quite stable. The complex derived from molecule ND_nw_193 was stable in NiV-G and NiV-N and plateaued to an average RMSD similar to that of RASE0125 and CARS0358. Apart from whole-molecule RMSDs, RMSDs of bound molecules alone were also computed, which also presented stable conformations (Figure 5). RMSF, a measure of the displacement of atom(s) relative to the reference structure, is another measure of structure assessment that captures local changes in protein structures. As a control, Cα RMSF values of all free proteins were also computed. Cα RMSF values indicated that the molecules RASE0125 and CARS0358 did not cause much fluctuations in the amino acids in NiV-G and NiV-N, indicating these to be not disturbed during ligand binding (Figure 5). Molecule ND_nw_193 also did not cause fluctuations in NiV-G and NiV-N, but the overall flexibility of the NiV-F protein structure increased upon ligand binding; these results are also in concordance with the results of RMSDs. Importantly, none of the critical binding-site residues showed large flexibility. The high peaks in the RMSF plots indicate fluctuations in loop regions that are comparatively higher than those in the structured regions due to their inherent structural flexibility. 

### 3.8. Details of Selected Multitarget Molecules

Overall, the molecules RASE0125 and CARS0358 were well capable of binding to NiV-G, NiV-F, and NiV-N proteins, whereas ND_nw_193 could only bind well to NiV-G and NiV-N. Both RASE0125 (17-O-Acetyl-nortetraphyllicine) and CARS0358 are indole alkaloids (phytochemicals), derived from plant *Rauvolfia serpentina* and *Catharanthus roseus*, respectively. Indole alkaloid derivatives are known to inhibit dengue and Zika virus infection by modulating the virus replication complex [68]. Therefore, we also speculate that indole alkaloids RASE0125 and CARS0358 might possess the same mode of action on NiV. Literature search confirmed that these molecules have never been considered even for antiviral drug discovery. Contrarily, ND_nw_193 (CHEMBL223402) is a chemical drug RSV604, which is a known inhibitor of human respiratory syncytial virus (RSV) [69].

RASE0125 and CARS0358 established many hydrophobic contacts and H-bonds with binding-site residues with all three viral protein targets. In NiV-G, RASE0125 established three hydrophobic contacts with Gln559, whereas CARS0358 established six hydrophobic contacts with Gln559, and ND_nw_193 established five hydrophobic contacts with Gln559. It is interesting to note that Gln559 is the common amino acid residue in NiV-G that binds to all three molecules. In NiV-N, RASE0125, CARS0358, and ND_nw_193 established 27, 40, and 34 hydrophobic contacts with binding-site residues (Appendix A), having residues Ile31, Val58, Leu128, Phe38, and Arg36 in common. In NiV-F, RASE0125 and CARS0358 established 28 and 22 hydrophobic contacts with binding-site residues (Appendix A), having residues Tyr30, His29, Lys40, Val39, Tyr432, and Leu433 in common. The common binding-site residues could also be of prime interest to virologists for anti-NiV drug discovery.

### 3.9. Principal Component Analysis and Gibbs Free Energy Calculation

To explore the conformational changes docked molecules (RASE0125, ND_nw_193, and CARS0358) brought into the viral protein structures, PCA was carried out utilizing the essential dynamics approach. The high fluctuations in residues of proteins are captured through PCA [70], while variation in GFE values is computed to assess protein stability [71,72]. Proteins regulate their functions via entering into different conformations. The overall conformational change is governed by the collective movements of the atoms in a protein and this internal motion can be measured using PCA analysis. Therefore, to study the collective motion of three selected molecules before and after binding with viral protein targets occupied in the conformational subspace during the simulation, PC1 and PC2 were computed. FELs deliver a precise portrayal of a protein’s most stable conformational ensembles, which are certainly important for the study of conformational changes underlying protein−ligand interactions. The FEL plots were constructed and analyzed using the first two PCs (eigenvectors). The corresponding free energy contour map with a deeper blue color indicates lower energy (global minima) and energetically favored protein conformations, and yellow spots reflect unfavorable conformations [73]. As an example, we extracted the conformations of docked complexes CARS0358-NiV-G/NiV-F/NiV-N at different time points and superimposed the respective ligand conformations (Figure 6). In all of the complexes, minor differences in ligand conformations were observed at each time point corresponding to changes in energy basins during the course of the simulation. The FELs of free proteins and CARS0358-bound proteins (final conformations) are provided in Figure 6. The FELs of RASE0125 and ND_nw_193 complexes with respective viral proteins are provided in Appendix A. In all of the FEL plots, free and molecule-bound complexes had different patterns for the free energies, and protein structures were still able to attain comparable energetically and structurally stable conformations. FEL analyses suggested that the presence of small molecules affected the size and the position of the sampled energy basin to achieve stable equilibriums.

## 4. Conclusions

In this study, a sequential molecular docking and molecular dynamics-based approach was implemented to prospect for potential multitarget chemical/phytochemical inhibitor(s) against NiV by simultaneously targeting NiV-G, NiV-F, and NiV-N. The study identified phytochemical molecules RASE0125 (17-O-acetyl-nortetraphyllicine) and CARS0358 (NA), which are indole alkaloids, as distinct multitarget inhibitors of all three viral proteins, while chemical molecule ND_nw_193 (RSV604) was revealed to be an inhibitor of NiV-G and NiV-N.

## Figures and Tables

**Figure 1 microorganisms-10-01181-f001:**
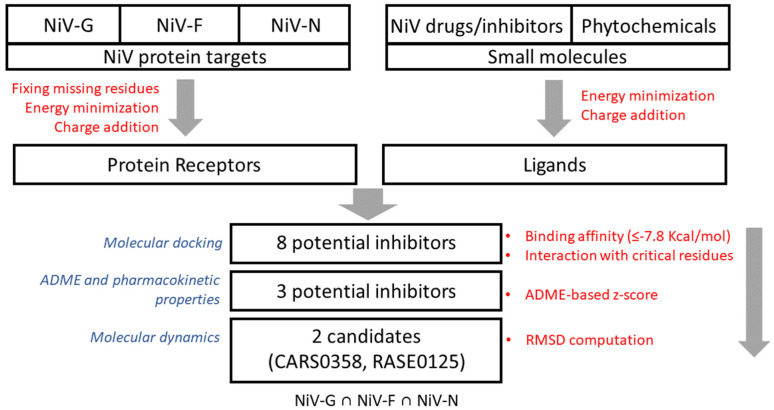
Workflow for the integrated approach for multitarget molecule screening in NiV.

**Figure 2 microorganisms-10-01181-f002:**
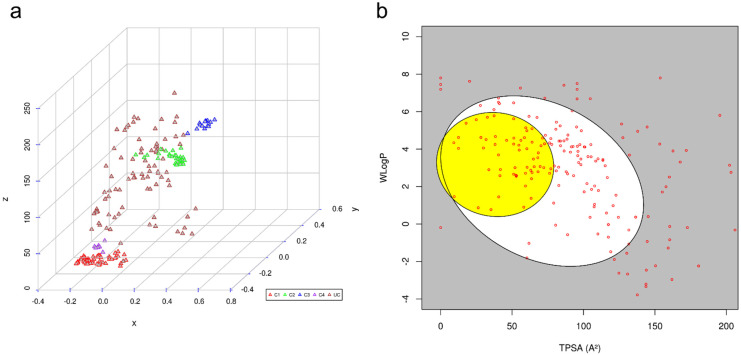
(**a**) Clustering results of molecules compiled from literature, where red-, green-, blue-, and purple-colored triangles represent Clusters 1–4. Unclustered molecules are represented as brown triangles; (**b**) graphical distribution of molecules compiled from literature as BOILED-Egg plot.

**Figure 3 microorganisms-10-01181-f003:**
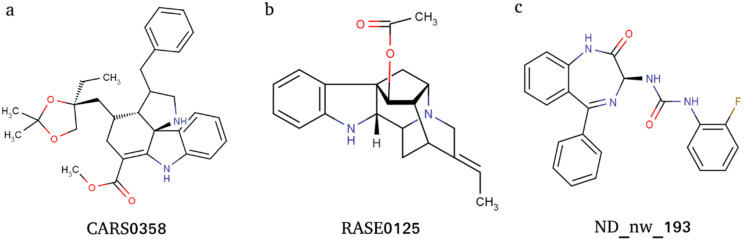
Two-dimensional (2D) images of molecules CARS0358 (**a**), RASE0125 (**b**), and ND_nw_193 (**c**).

**Figure 4 microorganisms-10-01181-f004:**
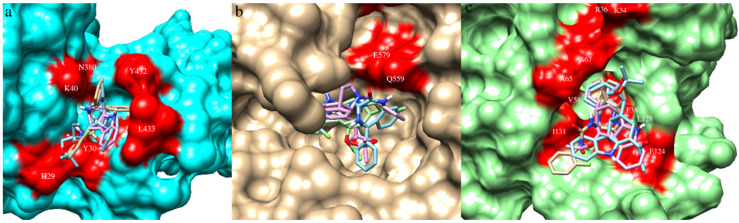
Molecule CARS0358 (blue), RASE0125 (pink), and ND_nw_193 (bisque) docked inside the respective ligand-binding sites (red colored) of NiV-F (**a**), NiV-G (**b**), and NiV-N (**c**).

**Figure 5 microorganisms-10-01181-f005:**
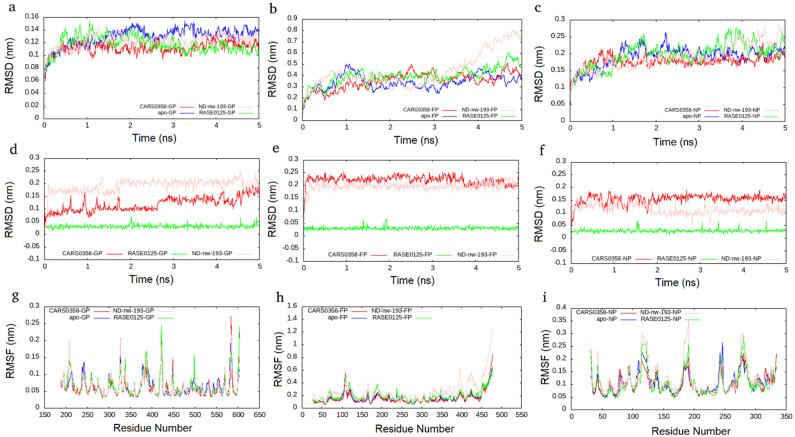
Cα−RMSD plots for selected molecules (CARS0358 (red); RASE0125 (green), ND_nw_193 (pink)), complexed with NiV−G (**a**), NiV-F (**b**) and NiV-N (**c**). Cα-RMSD plots for free proteins (blue) are also presented. RMSD plots for individual molecules in NiV−G (**d**), NiV−F (**e**), and NiV−N (**f**). Cα−RMSF plot for NiV−G (**g**), NiV−F (**h**), and NiV−N (**i**) as a function of time.

**Figure 6 microorganisms-10-01181-f006:**
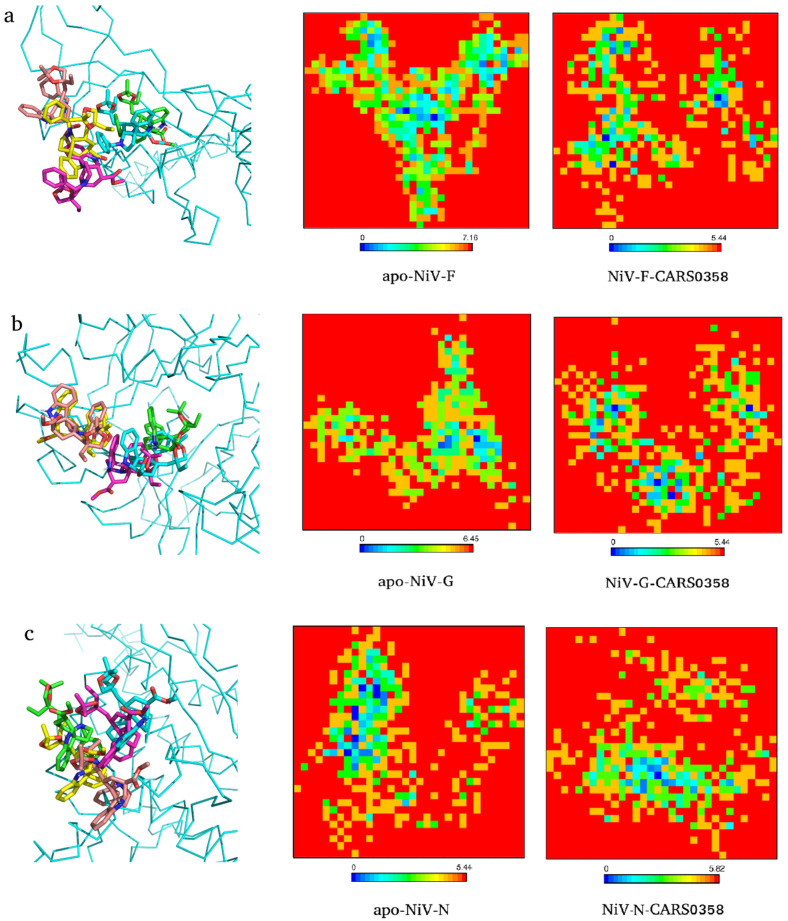
Conformations (1 ns (green), 2 ns (cyan), 3 ns (pink), 4 ns (yellow), and 5 ns (brick)) of molecule CARS0358 inside binding pockets of NiV-F (**a**), NiV-G (**b**), and NiV-N (**c**) during whole MD simulations (left panel). The respective 2D graphs of the Gibbs free energy landscape for apo- (middle panel) and final ligand-bound conformation (right panel) are also provided.

**Table 1 microorganisms-10-01181-t001:** Binding affinity (BA, Kcal/mol) values of 8 selected molecules against viral protein targets.

Mol. ID	Mol./Drug Name	Mol. Type	BA (NiV-F)	BA (NiV-G)	BA (NiV-N)
ND_nw_193	RSV604	Chemical	−8.6	−8.8	−9.0
ND_nw_93	AC1MH6FW	Chemical	−8.2	−8.3	−8.8
CARS0358	NA	Phytochemical	−7.9	−8.0	−8.4
CARS0394	NA	Phytochemical	−7.8	−7.8	−8.9
CARS0456	24-Olefinic sterol	Phytochemical	−7.8	−8.2	−8.0
pdtdblig00047	Naringin	Phytochemical	−8	−7.9	−8.4
RASE0125	17-O-Acetyl-nortetraphyllicine	Phytochemical	−7.8	−7.8	−8.6

**Table 2 microorganisms-10-01181-t002:** ADME-based z-scores for each molecule. Bolds are the top-ranked molecules selected for molecular dynamics simulations.

Mol. ID	Mol./Drug Name	Mol. Type
**RASE0125**	**17-O-Acetyl-nortetraphyllicine**	**1.55**
**ND_nw_193**	**RSV604**	**1.05**
**CARS0358**	**NA**	**0.79**
CARS0456	24-Olefinic sterol	−1.27
pdtdblig00040	Procyanidin B2	−0.73
CARS0394	NA	−0.73
ND_nw_93	AC1MH6FW	−0.48
pdtdblig00047	Naringin	−0.22

## Data Availability

Not applicable.

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
