# Peer review of "Computational Identification of Potential Multitarget Inhibitors of Nipah Virus by Molecular Docking and Molecular Dynamics"

_microorganisms, 2022, doi:10.3390/microorganisms10061181_

Round 1
Reviewer 1 Report
Randhawa and colleagues present a computational research for the identification of dual NiV glycoproteins G and F inhibitors based on compounds identified as Nipah drugs or inhibitors in PubMed.
The study was well-planned and executed, and the publication was well-written.
Only minor revisions are required, and the authors are encouraged to consider the following:
- There are a few grammatical and typographical issues that should be fixed.
- Nipah is a POTENTIALLY pandemic disease, not a pandemic disease. (30th line)
- In line 40, type the acronym HeV.
Reviewer 2 Report
The manuscript by Vinay Randhawa et al. would be very interesting, if it will be revised and better detailed. The applied computational protocol seems to be well-performed but it would be more reliable and robust by means of a specific validation protocol.
Main concerns about this work involve:
- poorly detailed methods about binding site analysis and docking calculation. concerning this issue, available data , if any, along the literature, should be cited.
- Docking of known inhibitors would be reported and discussed, even in comparison with the newly proposed derivatives.
- Docking results description is quite confusing and vague, please focus on the most relevant achieved information
- molecular dynamic simulation would be performed on reference inhibitors prior to the newly developed compounds.
Minor comments:
Introduction- please, insert a scheme about the compounds so far explored to contrast NiV infection, those previously developed by the research group and a flowchart with the applied protocol herein described.
Figure 3 is difficult to read, please revise it
Reviewer 3 Report
Vinay Randhawa et al, have conducted a pertinent Computational study on the identification of a potential entry inhibitor of Nipah virus glycoprotein G (NiV-G) and fusion 2 glycoprotein (NiV-F). The data was displayed nicely in easy to understand charts, and the topic is certainly relevant. However, in order to be accepted, it is in need of minor revisions. The results section is confusing and needs to be restructured.
Reviewer 4 Report
Apr. 10, 2022
Review on “Computational identification of a potential entry inhibitor of Nipah virus glycoprotein G (NiV-G) and fusion 2 glycoprotein (NiV-F) by molecular docking and molecular dynamics” by Vinay Randhawa, Shivalika Pathania and Manoj Kumar
The authors identified a small molecule inhibitor (molecule_181) simultaneously acting to NiV-G and NiV-F, which are responsible for NiV entry into host, using bioinformatics tools as well as physics-based simulations. Considering that Nipah virus (NiV) has brought sever damages in Malaysia and other Asian countries and is classified as biosafety level 4 (BSL4), the computational design of potential drugs is of importance. They started with known NiV inhibitors, and through systematic evaluations, proposed new multi-target drug candidate. This work provides an effective protocol for searching multi-target drug candidates. General concern about in-silico drug screening is the lack of any experimental validation. In case this is difficult, they should rationalize the result under the through comparison with some known inhibitors or characterizing the mechanism. For example, why does this molecule similarly bind to NiV-G and NiV-F which have electrostatically opposite binding pocket? Is there any example showing the similar binding? This work can be published, but the authors are encouraged to convince their results before the publication.
Comments and suggestions:
- The authors may benefit from adding one figure showing a flowchart of the processes to come down with "molecule_181".
- P2, L53: please correct “search search”
- P3, L100: What are the sizes of the largest biding pocket for NiV-G and NiV-F? The size of the second largest pocket for each protein may also be given to convince the current choice is reliable. In addition, Fig. 2 may be modified to show the position of each pocket with key binding site residues, Ile588 and Tyr581 for NiV-G and Val65 for NiV-F for better understanding.
- P3, L106: “appropriate charges” should also be explained precisely for reproducibility.
- P3, L115: “…were analyzed…” sounds better than “…were computed…”.
- P3, L138: The appropriate citation should be added for CHARMM36 force filed v2019.
- P3, L141: “the simple point charge (spc216)” may be changed to “the simple point charge water model (spc216)”
- P6, L23: How did the authors extracted four molecules? Was some consensus score used?
- P6, Table1: The properties of the least four molecules may also be listed for comparison.
- P9, L314: How do PC1 and PC2 vectors look like? What is the contribution of each of two vectors?
- P9, Fig. 5: Are these RMSD values for only protein Ca atoms? Please give the explanation in detail. To assess the stability of binding poses, RMSD for the bound ligand may also be shown and discussed. In addition, I wonder why in Fig. 5b two RMDSs (apo and bound forms) show very similar trajectories. Is this due to the short simulation time?
- P10, Fig. 6a: Please check if the PC modes between the apo and bound states correspond to each other. If the free-energy maps are drawn along different modes, the difference between two maps may not reflect the effect of binding.

Round 2
Reviewer 4 Report
Apr. 26, 2022
Review on “Computational identification of a potential entry inhibitor of Nipah virus glycoprotein G (NiV-G) and fusion 2 glycoprotein (NiV-F) by molecular docking and molecular dynamics” by Vinay Randhawa, Shivalika Pathania and Manoj Kumar
The authors have mostly addressed the points that I made. I think this work is publishable. I suggest just one point about the free-energy landscape (FEL). FLE analysis should be based on the sufficient sampling in its nature. The presented FEL is based on very short simulation (5 ns) and obviously not converged. The data can be presented to illustrate the possibility of multiple bound poses with some snapshots, but mainly I suggest the authors not to over-interpret the results. One idea is to leave only NiV-F/NiV-F-CARS0358 case as the main Fig.6 with some snapshots for three basins, and move the rest to SI.
No further review is needed.
Minor points:
- There is "C-alpha" which has not been changed to Greek letters.
- The unit of time axis in Fig. 5 should be “ns”? Please enlarge the font size of labels and ticks for better visibility.
- Please emphasize the residue name (white letters on the figure) in Fig. 4 for better visibility.
- KBT should be kBT
